# Mapping-by-Sequencing via MutMap Identifies a Mutation in *ZmCLE7* Underlying Fasciation in a Newly Developed EMS Mutant Population in an Elite Tropical Maize Inbred

**DOI:** 10.3390/genes11030281

**Published:** 2020-03-06

**Authors:** Quan Hong Tran, Ngoc Hong Bui, Christian Kappel, Nga Thi Ngoc Dau, Loan Thi Nguyen, Thuy Thi Tran, Tran Dang Khanh, Khuat Huu Trung, Michael Lenhard, Son Lang Vi

**Affiliations:** 1Department of Genetic Engineering, The Agricultural Genetics Institute, Km2 Pham Van Dong Street, Hanoi, Vietnam; 2Institute for Biochemistry and Biology, University of Potsdam, Karl-Liebknecht-Straße 24–25, House 26, 14476 Potsdam-Golm, Germany; 3Center for Expert, Vietnam National University of Agriculture, Trau Quy, Gia Lam, Hanoi, Vietnam

**Keywords:** EMS, MutMap, mutagenesis, CLE7, tropical maize, fasciation, mapping

## Abstract

Induced point mutations are important genetic resources for their ability to create hypo- and hypermorphic alleles that are useful for understanding gene functions and breeding. However, such mutant populations have only been developed for a few temperate maize varieties, mainly B73 and W22, yet no tropical maize inbred lines have been mutagenized and made available to the public to date. We developed a novel Ethyl Methanesulfonate (EMS) induced mutation resource in maize comprising 2050 independent M2 mutant families in the elite tropical maize inbred ML10. By phenotypic screening, we showed that this population is of comparable quality with other mutagenized populations in maize. To illustrate the usefulness of this population for gene discovery, we performed rapid mapping-by-sequencing to clone a fasciated-ear mutant and identify a causal promoter deletion in *ZmCLE7 (CLE7)*. Our mapping procedure does not require crossing to an unrelated parent, thus is suitable for mapping subtle traits and ones affected by heterosis. This first EMS population in tropical maize is expected to be very useful for the maize research community. Also, the EMS mutagenesis and rapid mapping-by-sequencing pipeline described here illustrate the power of performing forward genetics in diverse maize germplasms of choice, which can lead to novel gene discovery due to divergent genetic backgrounds.

## 1. Introduction

Maize is an important staple crop worldwide used for food, feed and industrial products including biofuels [1]. The demand for maize has steadily risen in past decades and is expected to continue to increase, with maize demand predicted to double by 2050 [2,3]. Maize is not only of agronomic importance; it has also been a very important model organism for genetic studies. 

Forward genetics in maize has been very fruitful and there are several mutant populations in maize; however, most of these are in standard inbred backgrounds. Most induced mutants in maize were created by two methods: pollen mutagenesis with Ethyl Methanesulfonate (EMS) or crossing with maize lines that carry highly active transposons [4]. As a mutagen, EMS has the unique advantage that it can potentially create allelic series ranging from null, hypomorphic, hypermorphic to gain-of-function alleles, which can be very useful for studying gene function and applied breeding. For introducing point mutations, treating pollen with EMS has been the preferred method, because it is much more efficient than treating maize seeds or the use of radiation [4,5]. Using pollen mutagenesis, Prof. Gerald Neuffer has created a large number of mutants that have been used by researchers globally till now [6]. Several additional mutagenesis populations have been created: a TILLING population in B73 comprising 750 M1 plants [7], an indexed population in B73 that has been exome-sequenced comprising 1086 M1 plants [8], and other populations in W22 [7] and in the Iodent inbred PH207 [9]. By transposon mutagenesis, several large mutation populations have been created for reverse genetics (reviewed in [10]): the rescueMu created in A188 × B73 (HiII hybrid) [11]; the MTM Mu in B73 [12]; TUSC (Trait Utility System for Corn) in mixed background developed by the Pioneer Hi-Breed Company, the Mu population developed by Biogemma, UniformMu in W22 [13,14], the Photosynthetic Mutant Hunt population [15], and the ChinaMu in B73 [16]. A γ-irradiation mutant population in B73 was also created [17]. Despite the tremendous usefulness of these populations, in the South Asia tropical area, for example in Vietnam, the temperate standard maize inbreds (e.g., B73, W22, Mo17) grow very poorly, are highly susceptible to leaf disease, and have large anthesis-silking intervals making them very difficult to conduct genetics with. 

Diversifying the genetic resources can help identify novel genes, whose function is masked by the genetic background of the standard inbreds, and is essential for studying adaptive, abiotic and biotic-stress related traits. Tropical maize inbreds have proved to be very valuable germplasm resources for several important traits. For examples, two drought-tolerant genes were cloned from the tropical inbred CIMBL55 [18,19], a head smut resistance and other disease resistance genes were identified from a Thailand Suwan and other tropical maize varieties [20,21,22,23]. Many of the studies on exotic germplasm in maize exploit natural variation, for example using GWAS [24]. So far, no artificial mutant population has been made in a non-standard maize inbred to allow forward genetic studies directly on the exotic germplasms. 

The lack of materials is not the only issue; to successfully conduct forward genetic studies, one must consider standing challenges in mutant mapping/map-based cloning. First, maize has a large and complex genome (~2300 Mb) with lots of repetitive and hypervariable regions [25,26]. As such, the maize genome is several times larger than that of the two model plant species *Arabidopsis thaliana* (~135 Mb) and rice (~430 Mb). Second, heterosis is so strong and prevalent in maize that many mutant phenotypes cannot be scored accurately in F2 mapping populations of the original mutant outcrossed with an unrelated inbred. Subtle, quantitative, and physiology-based phenotypes such as kernel row number, plant height, root length, flowering time, nitrogen use efficiency, abiotic and biotic-stress related phenotypes, etc. require an isogenic background to correctly classify phenotypes. Third, background modifiers from the second parent, which are common in maize, can interfere with phenotype scoring. Therefore, in many cases several rounds of backcrossing, each of which can show different modifying effects, and careful pedigree/segregation analysis are required to create good mapping populations. This process is time-consuming and sometimes impossible for complex traits and genetics. 

To solve these difficulties, several approaches combining next generation sequencing and mapping populations generated in the same genetic background have been developed. These include crossing to the unmutagenized parent in the Mutmap method in rice [27] and in maize [28], or crossing to a normal looking plant of a different M2 family (the so called “evil twin” in Sorghum [29]) or even not crossing at all, but only sequencing M3 plants as in Mutmap + (rice) [30], or sequencing individual M2 plants combined with zygosity analysis via progeny in maize [9]. These approaches utilized only the segregating mutagen-induced variants between the two parents or siblings to map, and hence they all shared the limitation of relying on the low number of mutagen-induced variants. It is unclear in maize whether these so called modified versions of the Mutmap approach can robustly identify causal mutations given the predicted low number of mutagen-induced variants for such a big, complex genome, especially when using non-reference germplasm.

In this study, we sought to create a new mutant population resource and establish a simple and rapid mutant mapping approach to facilitate the forward genetic study in a non-standard maize variety. Using the EMS pollen-mutagenesis method, we have created a high-quality mutant population in an elite tropical inbred ML10, one of the inbred parents for a very popular maize hybrid in tropical South Asia. We have used a modified Mutmap pipeline in maize that is simple, cost-effective and does not require crossing to an unrelated inbred. Instead the mutant in question was crossed to a ML10 line that was mutagenized for two consecutive generations with the aim to introduce more segregating variants for mapping. Despite this, the mutation was mapped based solely on the EMS-induced mutations in the focal mutant relative to the unmutagenized background, suggesting that even simple backcrossing to the unmutagenized background should be sufficient. We have illustrated the usefulness of this population and our mapping-by-sequencing strategy by cloning a fasciation mutant and identifying a promoter deletion in *ZmCLE7*. 

## 2. Materials and Methods 

### 2.1. Mutagenesis, Phenotypic Screening and Plant Growth Conditions

ML10 is the maternal parent of the popular elite hybrid LVN10 in Vietnam [31]. ML10 was used to create the mutagenized population. The inbred was developed by selfing followed by sib-mating and was released in 1994. The pedigree is unclear, but historical records indicate that it was developed from a CP hybrid (Charoen Pokphand Group, C.P. Tower, 313 Silom Road, Bangrak, Bangkok 10500, Thailand). BL10 is the paternal parent of hybrid LVN10, also developed by selfing and maintained by sib-mating. B73 is the reference inbred strain and was obtained from Professor David Jackson, Cold Spring Harbor Laboratory. Maize was planted in two seasons in Ha Noi, Vietnam (map coordinate: 21°06’23.8” N 105°49’34.7” E): for the spring season, sowing was done in January and for the autumn season, sowing was done in August. These sowing times were optimized as shown in Table 1. The EMS pollen mutagenesis was carried out according to [5]. In brief, maize pollen was collected in the morning from tassel bags that had been put on the tassels overnight. To prepare the EMS stock solution, EMS (Sigma M0880-1G) was diluted to 1% (v/v) in mineral oil (Sigma 69794-500ML); the bottle was covered in aluminum foil, and mixed thoroughly by a magnetic stirrer overnight before the mutagenesis experiment. This stock solution was used on the next day for making the final EMS solutions in mineral oil. About 12−20 mL pollen was incubated with 70 mL of mineral oil containing EMS at the concentration stated in the results for 30 min. The pollen mixtures were then pipetted onto the silks. The pollinated ears were covered with shoot bags. The concentration of EMS of 0.1% (v/v) was used to generate a large population of over 6000 M1 seeds. The resulting M1 plants were then grown on the field and selfed to create M2 families. 40 seeds per M2 ear were sown and scored for visible phenotypes. Researchers can obtain seeds from this mutant population and/or arrange a screen for the specific phenotype in this population for research/breeding purpose via a Material Transfer Agreement by directly contacting the corresponding author S.L.V.

### 2.2. Generating E-ML10

E-ML10 plants (the “Evil-twin” ML10 or “Evil” ML10) were ML10 mutagenized for two successive generations with 0.1% EMS. The first round of EMS was the same as in the generation of the large mutant population in Section 2.1. In the second round, we selected healthy wild-type looking M1 plants, which did not exhibit any visible dominant/deleterious mutant phenotypes, pooled pollen from these plants, treated the pollen with 0.1% EMS and used it to pollinate the sibling M1 plants from which the pollen was harvested. The E-ML10 used for the cross with E1-9 to generate the mapping population was a wild-type looking individual grown from a mixture of seeds after the second round of EMS.

### 2.3. Mapping of E1-9

E1-9 was identified as an ear fasciation mutant in an M2 family. After first identification, this M2 family was re-grown; eight randomly selected M2 plants were selfed. The resulting M3 families were analyzed to select homozygous mutants. Three F2 mapping populations were made by crossing the M4 homozygous mutant E1-9 to a second parent (E-ML10, BL10 or B73) to make F1s; multiple F1 plants were selfed to generate F2 families. 

Four samples used for next generation sequencing were one individual unmutagenized ML10 wild-type plant (named ML10), a pool of eleven mutant plants from a M5 homozygous mutant family (named E1-9), a pool of 72 plants with a mutant phenotype (named F2-mutant-pool) and a pool of 69 plants with wild-type phenotype (named F2-WT-pool) from the F2 mapping population E1-9 × E-ML10. The F2 plants in each pool were derived from four selfed F1 plants. 

### 2.4. DNA Extraction 

For PCR, DNA samples were extracted from maize leaves using CTAB protocols [32], then kept at −20 °C for long term storage. For next generation sequencing, DNA was first extracted with CTAB, treated with RNAse A, cleaned up with Phenol:Chloroform, then cleaned up one more time with Plant DNA mini extraction kit (Thermo). The DNA quality was checked by Nanodrop and gel electrophoresis. 

### 2.5. Sequencing and Bioinformatic Analysis 

Extracted DNA samples were sent to Macrogen for library preparation with TruSeq DNA PCR-Free kit and sequenced using 151 bp paired-end Illumina platform. We obtained 185,173,048 fragments for the ML10 wild-type pool, 173,761,432 for the M5 mutant, 207,190,254 for the pool of F2 wild-type individuals, 235,111,914 for the pool of F2 mutant individuals. Reads were mapped to the B73 reference genome (*Zea mays*, Ensemble release 40, http://plants.ensembl.org) using bwa mem [33] and further processed using samtools [34]. Variant calling was performed using bcftools [35,36]. Data analyses were done using R (R Core Team, 2017. https://www.R-project.org/), plots were generated using R/lattice [37] (http://lmdvr.r-forge.r-project.org). Variant callings with quality scores below 500 were discarded. Allele frequencies (ratio of sample over reference B73) for F2 pools were plotted for variants fixed between the E1-9 mutant sample (allele frequency equal to 1) and the ML10 wild-type sample (allele frequency equal to 0) and having an allele frequency between 0.25 and 0.75 in the F2 wild-type pool sample. Loess smoothing for average allele frequencies was plotted to visualize more general tendencies. Coding sequences (CDS) for known fasciation genes were extracted manually using IGV [38]. Sequences were translated to peptides using transeq and then multiple sequences aligned using muscle [39].

Sequencing data are available at NCBI SRA under accession number PRJNA602200. 

### 2.6. PCR Analysis 

All primers used were listed in Appendix A. PCR was performed according to standard procedures. The PCR mix for a 20 µL reaction was: Taq 2X Mastermix (NEB): 10 µL; template DNA: 4 µL; primers (10 µM): 0.8 µL each, DMSO: 1 µL, deionised water to 20 µL total volume. For genotyping the *cle7* promoter deletion, the PCR thermal condition was 95 °C: 5 min; then 39 cycles of: 94 °C: 30 s, 59 °C: 30 s, 72 °C: 1 min; ending with 72 °C 10 min. For SSR analysis, the PCR thermal condition was 95 °C: 5 min; then 39 cycles of: 95 °C: 30 s, 55 °C: 40 s, 72 °C: 40 s; ending with 72 °C 10 min. 

## 3. Results

### 3.1. Optimization of EMS Protocol for the ML10 Inbred and Development of the Mutant Population

ML10 is the maternal parent of a very popular hybrid, LVN10, in the South East Asian region, e.g., Vietnam, Laos and Cambodia [31]. The inbred was developed in the 1990s by phenotypic selection and compatibility testing. Due to its popularity, high adaptability, high combining ability, excellent seed germination and the ability to robustly shed non-clumpy pollen even under harsh environmental conditions, we chose ML10 as the inbred background for pollen mutagenesis. 

We tested several EMS concentrations: 0.04%–0.06%–0.1%–0.185%. EMS treatment at 0.185% gave almost no seeds, while EMS treatment at 0.04, 0.06 and 0.1% gave seeds. M1 seeds from the 0.1% EMS experiment had a germination rate of approximately 50% compared to ML10 wild-type. As we wanted to make a population with as high a mutation density as possible, an EMS concentration of 0.1% was chosen to generate a large mutagenized population. Ears pollinated with 0.1% EMS-treated pollen contained on average 30 seeds/ear compared to approximately 220 seeds per ear in untreated plants (Figure 1A). This is also the concentration recommended for B73 in [5] and higher than the concentration used for B73 in [8] of 0.067%. We noticed that the weather condition during pollination is very important; in particular, partially sunny weather with temperatures from 28 to 32 °C and humidity from 50% to 70% seemed ideal, as otherwise the pollen got clumpy very quickly after harvest and had reduced viability. For the tropical conditions in Ha Noi, Vietnam, the best time for pollination to get this optimum weather is the first two weeks of April for the Spring season and from October the 12th to November the 12th for the Autumn season (see Methods). 

Using the optimized sowing time and EMS concentration of 0.1%, we mutagenized ML10 and obtained over 6000 M1 seeds. We sowed out M1 seeds, selfed the resulting M1 plants, and got 2050 M2 families. 1185 M2 families were phenotyped to find mutants, and results are summarized in Table 2 and Figure 1B–I. Numerous mutants were found, many of which have phenotypes similar to classical mutants in maize like *sh2*, *su1*, *ramosa*, *fea*, *liguleless*, *dwarf*, etc., [6] demonstrating that our EMS mutagenesis generated a large number of induced mutations. 

### 3.2. Development of a Mapping-by-Sequencing Method via Mutmap to Map the E1-9 Mutant

#### 3.2.1. F2 Mapping Population Generated with a Heavily Mutagenized ML10 (E-ML10)

A challenge for mapping mutations in maize is that when crossed to an unrelated parent to generate an F2 mapping population, mutant phenotypes may vary due to residual heterosis and segregating genetic modifiers. To circumvent this problem, we developed a heavily mutagenized ML10, named the E-ML10 for “Evil-twin” or “Evil” ML10, which underwent two generations of EMS pollen mutagenesis, and used it as an alternative parent to generate F2 mapping populations (Figure 2B). We reasoned that this E-ML10 will contain many EMS-induced SNPs compared to the original ML10 that can be used as molecular markers, abundant enough for fine mapping. Once an F2 population was made between E-ML10, and the mutant and phenotype scoring were successful, we used a similar method for mapping-by-sequencing as MutMap in rice [27] or BSA-Seq/MutMap in maize [28].

As the first pilot experiment to test the possibility of mapping without the need to cross to an unrelated inbred, we mapped the mild fasciation mutant E1-9 identified in our screen. The E1-9 mutant had its ear tip mildly fasciated and the ear had a higher number of kernel rows than wild-type (Figure 2A). Fasciation phenotypes can be subtle and difficult to score due to phenotypic suppression after crossing to an unrelated inbred; therefore, this phenotype was suitable to test the above mapping method. We crossed the homozygous M4 mutant to the E-ML10, and in the F2 derived from four selfed F1 plants we obtained 72 fasciated mutants from a total of 280 plants. This segregation ratio suggested that the fasciation phenotype in E1-9 was caused by a single recessive mutation. We sequenced pooled DNA from 72 fasciated mutants (F2-mutant pool) and 69 wild-type looking individuals (F2-WT pool) from this F2 mapping population. As the ML10 whole-genome sequence had not yet been available, we also sequenced one original unmutagenized ML10 individual (ML10), and a pool of eleven M5 homozygous E1-9 mutants (E1-9) to aid variant calling (Figure 2B). 

#### 3.2.2. Sequencing and Bioinformatic Analysis

Using the Illumina Hi-Seq paired-end sequencing platform, we obtained 52−71 Gb data per sample with > 20× coverage of the maize genome (2300 Mb) (Table 3). Reads were mapped to the B73 reference genome and variants were called.

To identify EMS-induced segregating variants in the F2 for mapping, we used only variants that were fixed between the M5 mutant (having an alternative allele frequency of 1) and the ML10 wild-type samples (having an alternative allele frequency of 0), and had an allele frequency between 0.25 and 0.75 in the F2 wild-type pool samples. This allele-frequency filter was used to exclude EMS-induced SNPs from E-ML10 that were only present in individual F1 plants from which part of the F2 was derived. Of note, because the F2 was derived from more than one selfed F1 plant, this filter means that we only based the mapping on EMS-induced variants present in the E1-9 line versus the unmutagenized ML10 and discarded other induced SNPs from E-ML10, as their expected frequencies could not be determined due to the pedigrees used (Figure 2B). We then plotted the allele frequencies of these filtered variants along the 10 chromosomes of maize to identify variants enriched in the F2-mutant pool, but not in the F2-wt pool (Figure 2C). This identified a broad region of distorted segregation in the mutant pool on chromosome 4, with the strongest signal at the very top of this chromosome. In addition, we observed clusters containing a very high number of variants on chromosomes 1, 2, 3, 5 and 6; these likely reflect residual heterozygosity in the starting ML10 material.

Similar to these observations, Klein et al. [28] also found that the B73 line used for their mutagenized population contained segments that differed from the B73 reference genome. Liang et al. [40] sequenced several B73 stocks from different laboratories and also reported the presences of clearly defined genomic blocks containing haplotypes that differ from the published B73 reference genome. Similarly, such residual heterozygosity in our ML10 material would explain why the E1-9 mutant contained genomic segments that differed from the E-ML10 used for crossing and the sequenced unmutagenized ML10 individuals. Thus, using our pipeline the E1-9 mutation was mapped to the top of chromosome 4. 

#### 3.2.3. Linkage Confirmation by PCR Analysis and Identification of Causal Mutation

To confirm the location of the mutation, bulk linkage analysis by SSR markers was performed in an independent F2 mapping population (E1-9 crossed with BL10). Results showed that the E1-9 mutation was linked to marker *phi021* in bin 4.03 on top of chromosome 4 (Figure 2D). These results showed that our map-by-sequencing approach allowed the rapid identification of the putative region harboring the causal mutation. 

To identify the causal mutation in this region, we further looked at the above-filtered SNPs that were homozygous in the F2-mutant-pool sample and found only one such SNP at position Chr4: 8,764,229 (C in ML10 and T in E1-9). However, this SNP was in the intergenic region, and was 3 kb downstream from the closest gene LOC103655072, which encodes for a *Pectinesterase Inhibitor 38*. Hence, this SNP was unlikely to be the causal mutation in E1-9. 

Therefore, we also checked the nucleotide sequences of candidate genes whose mutations are known to cause ear fasciation including *CT2* [41], *FEA2* [42], *TD1* [43], Zm*GB1* [44], *FEA3* [45], *FEA4* [46], and *CLE7* [47,48]. No differences were found in the coding region of these genes (Appendix A) between ML10 and E1-9. However, we found a fixed 376 bp deletion (Chr4: 8,337,361−8,337,738) in the promoter of *CLE7* in E1-9 and F2-mutant-pool, but not in ML10, and segregating in the F2-wt pool (Figure 3A,B, Appendix A).

*CLE7* lies in bin 4.02 on top of chromosome 4, very close to the region identified in our whole genome sequencing analysis. As CRISPR-Cas9 derived *cle7* maize mutants have recently been shown to cause ear fasciation [48], this makes the promoter deletion in *cle7* in our E1-9 mutant the prime candidate for the causal mutation. The *cle7-a1*, *cle7-a2* CRISPR-Cas9 derived alleles cause a deletion of one nucleotide at + 10 and + 11 after the start of translation, respectively, hence resulting in a frameshift after the third amino acid in the protein and causing a premature stop codon in these likely null-alleles. Our mutation is a 376bp deletion at position −2 relative to the ATG start codon, which is likely to be a strong, close-to-null mutant. The large promoter deletion explains why we did not find this variant in our bioinformatic analysis of the Mapping-by-Sequencing data, but only identified the closely linked marker SNP. Although the mechanism is unclear, EMS has been shown before to cause insertions/deletions [8,49].

To test for co-segregation of mutant phenotype and the promoter deletion in *CLE7*, we designed a PCR marker for the deletion (Figure 3B) and tested it in two different F2 populations. In one F2 (E1-9 crossed with B73), where a 1:3 segregation ratio of the mutant phenotype was observed (33 mutants: 95 wt plants), there was perfect co-segregation of the mutant phenotype with the homozygous deletion in all 33 mutant plants (Figure 3C). This result supports the hypothesis that the deletion in *CLE7* was the causal mutation in E1-9. In a different F2 (E1-9 crossed with BL10), we observed a much lower frequency of the mutant phenotype (18 mutants: 96 wt plants), significantly different from a 1:3 ratio (Chi-square test, *p* = 0.02). Although all of the 18 plants with the mutant phenotype were homozygous for the promoter deletion, confirming co-segregation, three plants, whose ears looked normal, were homozygous for the promoter deletion (Figure 3D). These three plants were genotypically mutant, but phenotypically wild-type, suggesting that the mutant phenotype was suppressed in these plants.

Taken together, these results strongly suggest that E1-9 was a *cle7* mutant; interestingly, they also suggest that there was a suppressor of the *cle7* mutant phenotype in the BL10 background.

## 4. Discussion and Conclusions

In this study, we successfully mutagenized an exotic elite tropical maize inbred by EMS treatment on pollen. We generated a 2050-family strong M2 population and identified numerous mutants with visible phenotypes. Many mutants identified in our population suggest the usefulness of this population as a complement to the B73 mutant collection. For example, B73 has yellow seeds while ML10 has orange seeds, and we identified several mutants that change the orange color of ML10 seeds to yellow and to white (Figure 1G–H). These mutants will be useful in identifying the natural variation that boosts provitamin A content in maize seeds [50,51]. Despite extensive mutagenesis screens, mutants in *CLE7* had not been found in any EMS populations of standard inbreds, but were identified in our population. Since our allele was created by EMS mutagenesis in an elite parent, it can directly be used in a breeding program to improve kernel row number (KRN) without transgenic regulation as compared to the CRISPR-Cas9 alleles. Our results suggest that there was a genetic suppressor(s) of the *cle7* phenotype in BL10, the paternal parent in the elite hybrid paired with maternal ML10. Combining this suppressor and *cle7* in ML10/BL10 hybrids could lead to an improved hybrid with a higher KRN and yield without being fasciated. Mapping of this suppressor could also identify a new factor controlling meristem proliferation. The presence of a suppressor of the *cle7* mutant phenotype in the BL10 background, but not in B73 emphasizes the value of using diversified non-reference genotypes in genetic studies, allowing the study of natural suppressors and enhancers.

Our mapping-by-sequencing method is similar to MutMap in rice [27]. Like Mutmap, our mapping procedure does not require crossing to an unrelated inbred, which makes scoring of phenotypes easier and more accurate. First introduced in rice with a genome size of 370 Mb, MutMap was successfully used to map several mutants using fewer than 2300 segregating EMS-induced variants. For maize with an 8× bigger genome and many more repetitive sequences, it was questionable whether this method would work, in particular when using exotic germplasm for which no reference genome is available. To our knowledge, our study is the second example of MutMap in maize where EMS-derived variants were used for mapping without the need to cross to an unrelated inbred. The first example was from [28], where pools of nine F2 mutants and of nine wild-type siblings were sequenced, as well as the unmutagenized parent of unknown genetic background. Mutants with subtle phenotypes or ones whose phenotype is affected by residual heterosis, which is very common in maize, will benefit most from this approach. Our bioinformatic pipeline quickly identified the chromosomal region carrying the mutation based on only 1118 segregating variants spread largely evenly across the 10 chromosomes and pools of 69 wild-type and 72 mutant F2 plants. The number of variants used in our mapping was two-fold less than in MutMap in rice, which is due to our more stringent variant filtering to eliminate the potential false variant callings caused by the complex structure of the maize genome and the low-frequency EMS-induced marker mutations introduced only via individual F1 plants. A low number of segregating variants can be a disadvantage because it will make peak calling difficult and limit the resolution of mapping; at the same time, it has the potential advantage of having only a few putative SNPs in the identified region to pinpoint the causal mutation. Our success in mapping by Mutmap in a non-standard maize inbred, where no prior genetic studies had been performed, will facilitate the adoption of this method to allow the rapid cloning of maize genes in diversified germplasm. 

As mentioned above, our analysis used only variants between ML10 and the M5 mutant. In principle, we could have used more variants from the more heavily mutagenized E-ML10 line; however, genomic complexity in the non-reference ML10 genome, the heterozygosity of E-ML10 for many of the induced mutations and the need to collect F2 samples from multiple F1 ears prevented us from doing so. In the future, a modification to the protocol will be to self E-ML10 several times to reduce heterozygosity of the induced marker mutations before crossing to the mutant of interest and to use plants from only one single selfed F1 plant for mapping. In addition, sequencing of the very parents from which the F1 plant(s) were derived (rather than sister plants from the parental generation) would be advisable.

Many mutations (for example weak *fea* mutants affecting kernel row number) in maize in B73 proved to be very difficult to map due to scoring F2 phenotypes in mixed genetic backgrounds (unpublished data from David Jackson). Based on our experience with mapping E1-9, we also generated a heavily mutagenized B73 line (the E-B73), and are now in the process of selfing it for several generations to fix the EMS-induced marker SNPs; this should result in an essentially isogenic second parent carrying a large number of SNP markers for mapping, which should be very useful to the maize research community.

In conclusion, we have developed an EMS population in an elite tropical inbred and found numerous interesting mutant phenotypes for forward genetic studies. We successfully identified a mutation in *CLE7* underlying ear fasciation by mapping-by-sequencing (MutMap). The EMS mutagenesis and rapid mapping-by-sequencing pipeline described here may encourage maize researchers to perform forward genetics in their maize germplasm of choice, which can lead to novel gene discovery due to diversified genetic backgrounds. 

## Figures and Tables

**Figure 1 genes-11-00281-f001:**
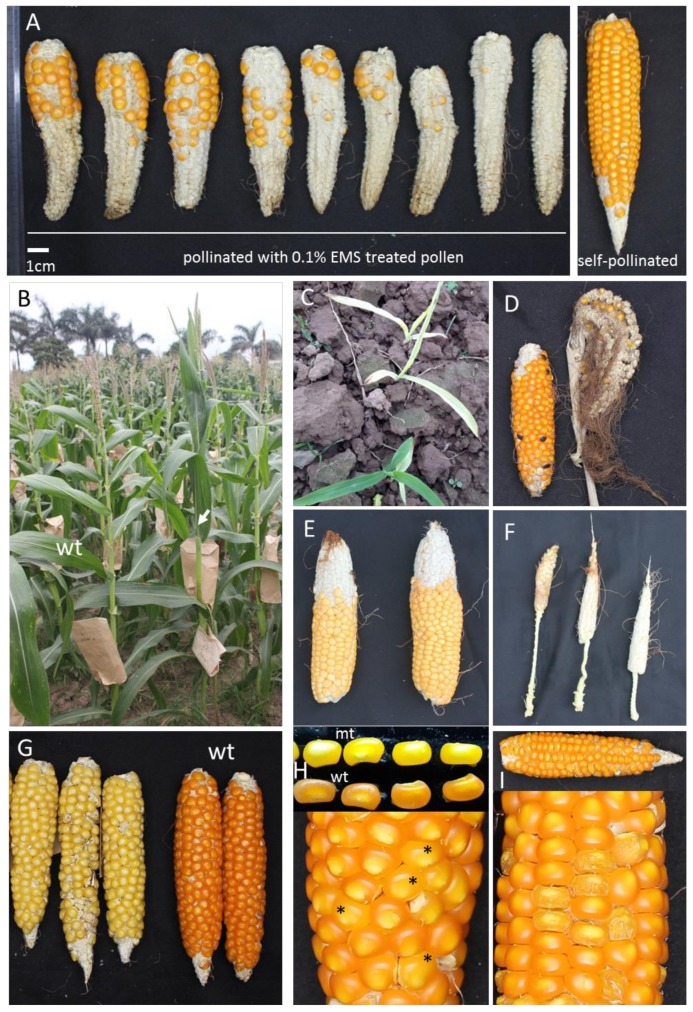
Typical M_0_ ears and mutant phenotypes observed in the M_1_, and later-generation mutagenized plants. (**A**) Typical M_0_ ears of ML10 that were pollinated with 0.1% EMS-treated pollen and a control self-pollinated ear. (**B–I**) Examples of mutant phenotypes observed. (**B**) A dominant M1 *liguleless* mutant with upright leaf angle (arrow) next to wild-type (wt) control. (**C**) Albino. (**D**) Tassel-seed (ear and tassel). (**E**) Fasciated ear. (**F**) Small ear with long shank. (**G**) White kernels; wt ears have orange kernels. (**H**) Yellow kernel mutant (mt) (kernels with asterisk *). (**I**) Wrinkled kernels *sugary*.

**Figure 2 genes-11-00281-f002:**
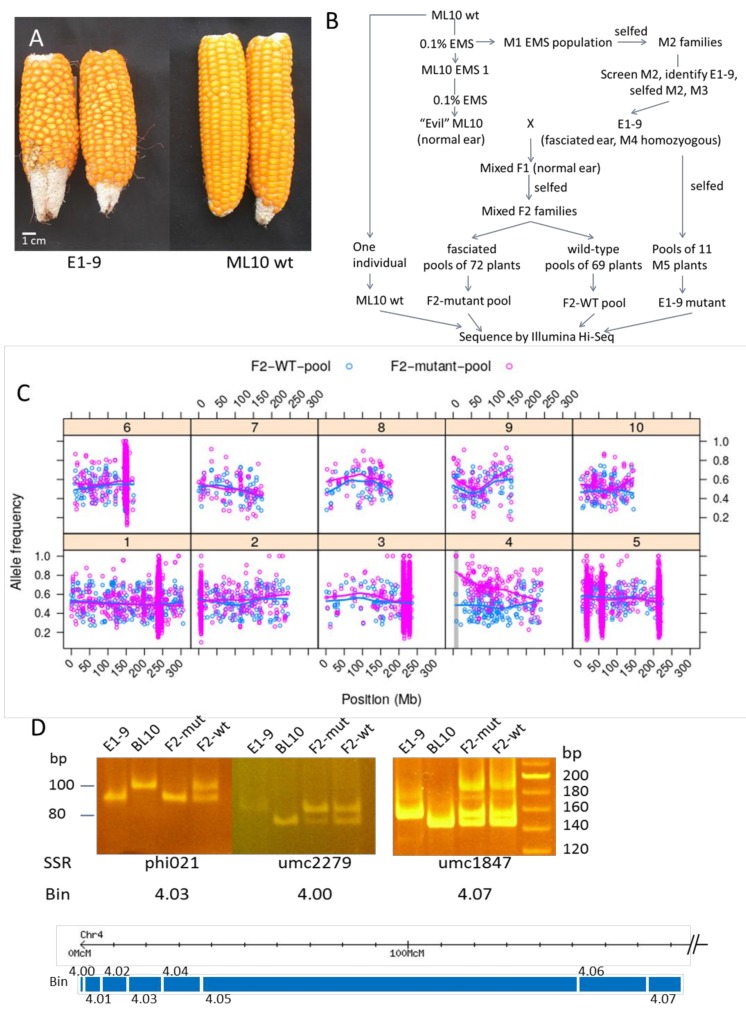
A modified Mutmap to map the E1-9 fasciation mutant by crossing with the “Evil” ML10. (**A**) Ears of E1-9 fasciation mutant and the wild-type ML10. (**B**) Crossing scheme and material preparation for bulk-sequencing. (**C**) Allele frequencies of filtered mutant vs wild-type markers for F2 mutant (purple) and F2 wild-type (blue) pools shown as dots. Lines represent Loess smoothing through allele frequency averages over 1 Mb windows. Vertical gray bar shows *CLE7* position. In the clusters with high variant densities on chromosomes 1, 2, 3, 5 and 6 the dots for the wild-type pools are obscured by the dots for the mutant pools. (**D**) SSR marker analysis in E1-9/BL10 F2 families confirms that the E1-9 mutation is linked to the top of chromosome 4. DNA samples are E1-9 and BL10 parents and their F2 pools of plants with mutant (F2-mut) and with wild-type phenotypes (F2-wt). Below is the genetic map of Bins on chromosome 4 (adapted from [6]). *CLE7* is in bin 4.02.

**Figure 3 genes-11-00281-f003:**
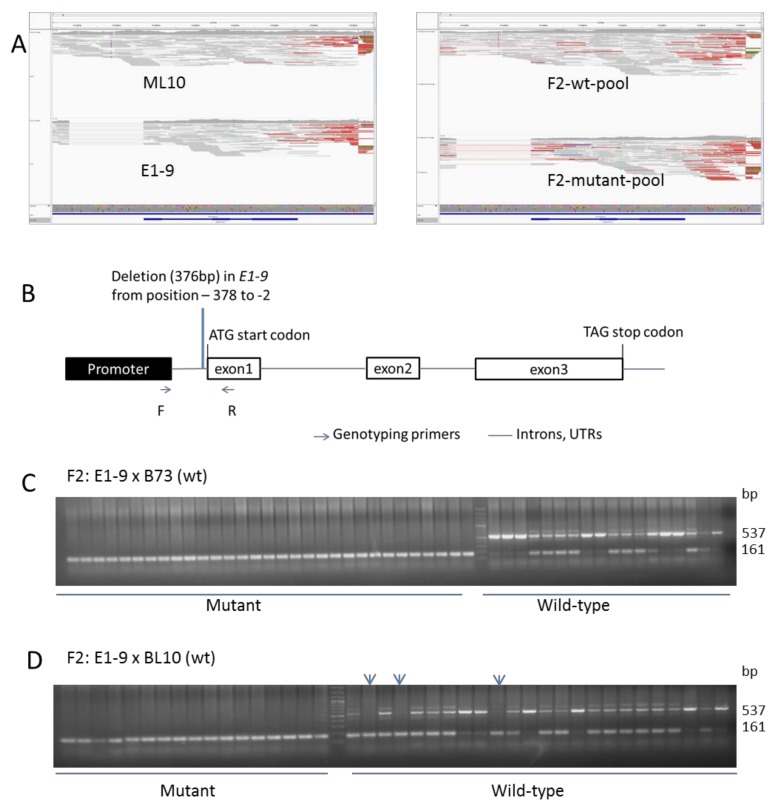
Identification of a promoter deletion in *CLE7* in E1-9. (**A**) IGV screenshot showing the alignments of reads to the B73 reference genome for ML10, E1-9, F2-wt pool and F2-mutant pool at the *CLE7* locus. (**B**) Illustration of the *CLE7* gene and the position of the promoter deletion in the E1-9 mutant. Genotyping primers were designed such that the wild-type allele gives a product of 537 bp, while the mutant allele gives a product of 161 bp. (**C**) and (**D**) Genotyping results for the phenotypically mutant and wild-type plants from two F2 populations made from crosses between the homozygous E1-9 mutant to B73 (**C**) and to BL10 (**D**), respectively. (**D**) In the F2 with BL10, arrows point at samples from plants with wild-type looking ears that had mutant genotypes, suggesting that BL10 contains a suppressor(s) of the *cle7* mutant phenotype.

**Table 1 genes-11-00281-t001:** Optimized sowing date for EMS mutagenesis in ML10.

Season	Sowing Date	Pollen Shedding Date	Estimated Time from Sowing to Shedding
Spring	Jan 11 (±10 d)	Apr 5 (±10 d)	85 days
Autumn	Aug 8 to Sep 8	Oct 12 to Nov 12	65 days

**Table 2 genes-11-00281-t002:** Frequency of typical mutants observed among 1185 M2 families screened.

Phenotype	Number of M2 Families	Frequency (%)
Defective kernel	33	2.78
Small kernel	7	0.59
Dwarf plants	4	0.34
Kernel color	4	0.34
Narrow leaf	4	0.34
Albino	3	0.25
Purple stalk/leaf	3	0.25
Fasciated ear	2	0.17
Chlorotic lesion leaf	2	0.17
Liguleless/upright leaf	2	0.17
Wrinkled kernel	3	0.25
Tassel branch angle	2	0.17
Tassel branch number	2	0.17
Tassel seed	2	0.17
Branched ear (*ramosa*)	1	0.08
Anther color	1	0.08
Broad leaf	1	0.08
Early senescence	1	0.08

**Table 3 genes-11-00281-t003:** Sequencing data summary.

Sample	F2 Mutant Pool	F2 Wild-Type Pool	E1-9	ML10
Number of reads	470,223,828	414,380,508	347,522,864	370,346,096
Number of mapped reads	462,796,855	407,731,780	344,923,846	361,795,129
% of mapped reads	98.42%	98.40%	99.25%	97.69%
Properly paired reads	405,367,374	359,813,078	305,113,228	320,163,016
% of properly paired reads	86.21%	86.83%	87.80%	86.45%
Coverage (x)	26	23	19	21

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
