# Peer review of "Mapping-by-Sequencing via MutMap Identifies a Mutation in ZmCLE7 Underlying Fasciation in a Newly Developed EMS Mutant Population in an Elite Tropical Maize Inbred"

_genes, 2020, doi:10.3390/genes11030281_

Round 1
Reviewer 1 Report
The manuscript “Mapping-by-sequencing via MutMap Identifies a Mutation in ZmCLE7 Underlying Fasciation in a Newly Developed EMS Mutant Population in an Elite Tropical Maize Inbred” by Tran et al. describes an EMS population of tropical inbred maize and shows the identification of putative ZmCLE7 allele as an example. The purpose of the study is clearly described and well justified in introduction. The size of mutant population and successful identifying mutant allele indicate the fineness of the new mutant population. Authors also showed successful application of MutMap method in the newly generated mutant population.
Some mention about how this mutant population can be accessible to the researchers would be helpful.
Do you have data showing how many SNPs were generated after two round of EMS treatment in your E-ML10 line? Do you think more rounds of EMS treatment would be helpful for your MutMap strategy? Please discuss about it.
L115 Providing exact geographical location of the field in Ha Noi would be helpful
L406 3.
Reviewer 2 Report
Brief summary:
Quan Hong Tran et al. 2020 developed an EMS pollen mutagenized maize population in an elite tropical inbred ML10. The authors appropriately described their mutagenesis procedure and characterized the EMS population by describing interesting mutant phenotypes. For one example, a fasciating ear phenotype, the authors successfully applied a MutMap approach and mapped the underlying mutation, a deletion in the promoter region of the ZmCLE7 gene. They furthermore generated two independent mapping populations by crossing their mutant E1-9 with B73 and BL10, on which they could demonstrate the co-segregation of the promoter deletion with the mutant phenotype.
Broad comments:
Overall, the manuscript is of interest to the reader and the scientific methods are appropriately described.
The different parts of the manuscript (e.g. introduction, M&M, results, and discussion) strongly differ in regard to quality. While the introduction and the discussion are well written, some chapters in the material and methods and results part need style and grammar checking.
It is necessary that the authors precisely differentiate between a mutant plant, a wild-type plant, a mutant plant with a mutant phenotype, and a mutant plant with a wild-type phenotype.
It is a very strong argument that the authors invested time and effort to generate two additional mapping population in which they independently found the co-segregation of the promoter deletion with the mutant phenotype.
Specific comments:
Line 46-47: The authors claim that EMS pollen mutagenized M1 plants are completely heterozygous rather than chimeric. It would be necessary to support this claim with a citation. Otherwise it is doubtful as maize pollen is in the tricellular G1 stage which means that the replication would start after fertilization. Pollen mutagenesis could result in differently mutated DNA strands, see [9] Heuermann et al., 2019; Figure S4.
Line 92-95: The authors doubt the efficiency of modified MutMap versions because of the low number of mutagen-induced variants, but prove themselves wrong by successfully mapping their mutation relying only on the EMS-induced variants between E1-9 and ML10. Furthermore they claim that the required sequencing coverage is unknown, which is doubtful as it is minimally defined by the pool size. If one pools DNA from x number of mutant plants the allelic state of a variant would only be precisely determinable by sequencing every allele at least once, x number of time, assuming perfect homogeneous distribution of DNA molecules. A lower coverage than the size of the pool is therefore an approximation.
Line 110-111: If the ML10 genotype was characterized somewhere else a citation would be helpful here.
Line 118-119: For the purpose of repeatability by other researcher, the exact molarity of the used EMS or the precise product (Sigma) used for the experiment should be reported.
Line 130: It is necessary to specify what “two rounds” means. I assume it means two generations?
Line 135: “Several M2 plants were selfed…” why and how were those plants chosen?
Line 143: How many F1 plants were used to generate the F2 plants which were sampled for DNA pools?
Line 211: Figure 1 (H), It is possible but hard to distinguish between yellow and orange kernels. Maybe the authors have another image in which the differences are more pronounced.
Line 271-273: Did the authors use different stocks of ML10 for their experiments or did they observe the residual heterozygosity inside the same stock.
Line 278-280: The authors identified a broad region of distorted segregation on chromosome 4 in their E1-9 x E-ML10 population. Why didn’t they also show their SSR marker analysis in the very same population to complement their mapping but only in an independent E1-9 x BL10 population?
Line 299: Wouldn’t it be possible to detect an expression increase, by simple RT-qPCR, between the E1-9 and the ML10 like [47] Rodriguez-Leal et al., 2019 found in their mutants to support the claim of a null mutant?
Personal remark: The authors’ arguments for the mapped mutation are strong enough and such an experiment is not necessary to support their claim.
Line 336: In the legend of Figure 3 (D) the authors claim that the presence of a suppressor is suggested by their finding. Thus, I would rephrase the sentence to “Our results suggest that there was a genetic suppressor…”) instead of “showed”.
